# Comparison of Lineblot and Immunoprecipitation Methods in the Detection of Myositis-Specific and Myositis-Associated Antibodies in Patients with Idiopathic Inflammatory Myopathies: Consistency with Clinical Diagnoses

**DOI:** 10.3390/diagnostics14192192

**Published:** 2024-09-30

**Authors:** Fabrizio Angeli, Eleonora Pedretti, Emirena Garrafa, Micaela Fredi, Angela Ceribelli, Franco Franceschini, Ilaria Cavazzana

**Affiliations:** 1Rheumatology and Clinical Immunology Unit, ERN ReCONNET Centre, ASST Spedali Civili, Piazzale Spedali Civili, 1, 25124 Brescia, Italy; fabrizio.angeli02@gmail.com (F.A.); eleonorapedretti@gmail.com (E.P.); micaela.fredi@unibs.it (M.F.); franco.franceschini@unibs.it (F.F.); ilariacava@virgilio.it (I.C.); 2Clinical and Experimental Science Department, University of Brescia, Piazzale Spedali Civili 1, 25124 Brescia, Italy; 3Central Laboratory Unit, ASST Spedali Civili, Molecular and Transitional Medicine, University of Brescia, Piazzale Spedali Civili 1, 25124 Brescia, Italy; emirena.garrafa@unibs.it; 4Rheumatology and Clinical Immunology IRCCS, Humanitas Research Hospital, via Manzoni 56, Rozzano,20089 Milano, Italy; 5Department of Biomedical Sciences, Humanitas University, Pieve Emanuele, 20072 Milano, Italy

**Keywords:** idiopathic inflammatory myopathies, dermatomyositis, polymyositis, immunoprecipitation, lineblot assay, autoantibodies

## Abstract

**Background:** the reference method for detection of myositis-specific and myositis-associated antibodies (MSAs and MAAs) is considered immunoprecipitation (IP), but it is routinely replaced by semi-automated methods, like lineblot (LB). Few data are available on the consistency with clinical diagnoses; thus, we aim at analysing these aspects. **Methods:** sixty-nine patients with idiopathic inflammatory myopathies (IIM) were studied via LB (Myositis Antigens Profile 3 EUROLINE, Euroimmun) and IP (RNA and protein antigens). The degree of concordance between methods was calculated using Cohen’s coefficient. **Results:** a substantial concordance was found for anti-Ku and anti-PM/Scl and a moderate concordance was found for anti-Jo1 and anti–Mi-2, while a fair concordance was found for anti-EJ, anti-SRP, and anti-Ro52 antibodies. The concordance could not be calculated for anti-OJ, anti-PL-7, anti-PL-12, anti-NXP2, anti-TIF1ɣ, and anti-MDA5, because they were only detected with one method. Multiple MSAs were found only with LB in 2/69 sera. Anti-MDA5, TIF1ɣ, NXP2 (detected via IP), and anti-Jo1 in anti-synthetase syndrome (both LB and IP) had the best concordance with clinical diagnosis. **Conclusions:** LB and IP show substantial concordance for PM/Scl and Ku, and moderate concordance for Jo1 and Mi-2, with a good concordance with clinical diagnoses. IP shows a high performance for DM-associated MSAs. LB seems to be more sensitive in detecting anti-Ro52 antibodies, but it identified multiple MSAs, unlike IP.

## 1. Introduction

Idiopathic inflammatory myopathies (IIMs) are a group of heterogeneous autoimmune inflammatory diseases with different phenotypes, primarily involving muscle and skin [1]. IIMs are currently classified based on clinical phenotype or histopathological features in dermatomyositis (DM), polymyositis (PM), anti-synthetase syndrome (ASyS), overlap myositis (OM), immune-mediated necrotising myopathy (IMNM), and sporadic inclusion-body myositis (IBM) [1].

Different autoantibodies directed against intracellular nuclear and cytoplasmic components have been detected in approximately 60–65% of IIM patients. These autoantibodies are divided in two groups, namely myositis-specific autoantibodies (MSAs), which are unique to IIM and usually mutually exclusive to one another, or myositis-associated autoantibodies (MAAs) which can occur also in other connective tissue diseases (CTDs) or overlap syndromes, but are not specific for myositis [2,3]. MAAs can also co-exist with other MSAs.

EULAR/ACR classification criteria for IIMs include only anti-Jo1antibodies as an immunological parameter [4], although several other MSAs are today considered important tools for identifying clinical subsets of IIMs and for defining patients’ prognosis [3,5,6]. Different established laboratory methods are today available for the detection of MSAs and MAAs, with different advantages and limitations, but the reference method for the detection of MAAs/MSAs is still considered immunoprecipitation (IP). However, IP has several limitations, and thus, it is performed in a few research settings worldwide: it requires staff with expertise in the techniques and in the interpretation of results; protein-IP requires the use of radioactivity in specific facilities; it is a method that requires several days before obtaining results [7,8]. For these limitations related to IP, different commercial immunoassays have been developed, in particular lineblot (LB) or dot blot, to try to replace IP in clinical practice, considering their low cost, rapidity in the detection of multiple MSA without a peculiar expertise, and their wide availability in non-research laboratories. However, concerns have been raised regarding the validity of these alternative assays, as false positive rates for MSAs are too high, as shown in one study in which 16% of healthy controls tested positive for MSAs [8,9], while other studies reported a low correlation for some MSAs when compared to IP, with a high frequency of multiple MSAs in the same sera [7].

The aims of our study are to analyse the concordance of a commercial LB and homemade IP assays in the detection of MSAs and MAAs in patients with a well-established diagnosis of IIM and to assess the consistency of the laboratory results with the clinical myositis phenotype.

## 2. Materials and Methods

Sixty-nine patients with known diagnoses of IIM, followed in our centre between 2010 and 2022, and with sera available for both LB and IP tests, were included in the analysis. Starting from the court of patients published in 2016 [8], an additional 11 sera from patients with IIM were analysed both with LB and IP. Diagnoses of previously published patients were re-evaluated and confirmed: DM, PM, and OM were defined according to Bohan and Peter Criteria [10] and revised according to ACR/EULAR criteria [4], whereas ASyS [11], IBM [12], and IMNM [13] were diagnosed according to the currently used definitions. A diagnosis of IIM in overlap with other systemic autoimmune diseases was made when recent criteria for systemic sclerosis (SSc) [14], Systemic Lupus Erythematosus (SLE) [15], Sjogren’s Syndrome (SS) [16], and rheumatoid arthritis (RA) [17] were met. Clinical data were obtained from clinical charts, and disease onset was defined as the first skin, joint, muscle, or constitutional symptom/sign related to IIM. For all the patients, demographic and epidemiological data, and extra muscular findings, including skin manifestations (heliotrope rash, Gottron’s papule, mechanic’s hands, sclerodactyly, and cutaneous ulcerations), calcinosis, arthritis, Raynaud’s phenomenon, dysphagia, and myocarditis, were collected. Muscle involvement was defined when patients presented at least one condition among the following: muscle enzymes’ elevation, muscle weakness, presence of typical electromyography (EMG) alterations, and/or inflammatory findings on muscle biopsy. Antinuclear antibodies (ANAs) were detected via indirect immunofluorescence (IIF) using HEp-2 cells (BioRad, Hercules, CA, USA).

Data related to MSA and/or MAA identification were collected from clinical charts, and MAAs were detected via commercial LB (Euroimmun Autoimmune Inflammatory Myopathies with 16 antigens: OJ, EJ, PL-12, PL-7, SRP, Jo1, Ro52, PM/Scl-75, PM/Scl-100, Ku, SAE1, NXP2, MDA5, TIF1-gamma, Mi-2alpha, Mi-2beta; Euroimmun, Lübeck, Germany), as routine practice. The same 69 sera were analysed with IP. Patients’ sera were isolated from whole blood through centrifugation at 2000× *g* for 15 min and then stored in a −20 °C freezer until use. Protein-IP was performed using ^35^S-methionine-labeled K562 cell extract, followed by SDS-PAGE and autoradiography, and via RNA-IP using unlabelled K562 cell extract followed by urea-PAGE and silver staining [18], as per established protocols [19,20].

This retrospective study was performed according to the principles of the Declaration of Helsinki.

Categorical variables were expressed as a number or percentage, and continuous variables were expressed as the median and interquartile range (IQR). Comparisons between groups were performed using the chi-square test, Fisher’s exact test, Student’s *t* test, and Mann–Whitney test when appropriate. A *p*-value < 0.05 was considered statistically significant. The degree of concordance between the LB-positive autoantibodies and IP-positive autoantibodies for patients was calculated using Cohen’s coefficient according to the Landis and Koch ranking: slight (k 0.01–0.2), fair (k 0.21–0.40), moderate (0.41–0.6), substantial (0.61–0.80), and almost perfect (0.8–1). This study was approved by the Ethical Committee of the leading centre (ASST Spedali Civili of Brescia, NP3511).

## 3. Results

Clinical diagnoses were distributed as follows: 11 ASyS (15.9%), 27 DM (39.1%), 23 PM (33.3%), and 8 (16.3%) overlap syndrome. In the DM cohort we included a patient with juvenile dermatomyositis (jDM), diagnosed at the age of 7. A total of 51 patients were female (73.9%), and 18 were male (26.1%). Most of the patients were Caucasian (66 patients, 95.6%). Demographic data are reported in Table 1. No significant differences were detected between different IIM subtypes. ANA results were available for 65/69 (94.2%) patients, with the titre for only 52/69 (75.3%), with a prevalent speckled pattern found in 36/65 (55.3%) samples, followed by cytoplasmic 9/65 (13.8%) and homogeneous 5/65 (7.6%) patterns.

### 3.1. Autoantibodies Detected via LB

Using the LB method, MSAs were detected in 25 (36%) sera, and only two patients had more than one MSA (2.8%). The most frequent MSA found was anti-Jo1, detected in 11 sera (16%), followed by non-Jo1 anti-synthetase autoantibodies (ASAs) found in 7 sera (10%). In particular, three patients were anti-PL-7 (4%), two were anti-PL-12 (3%), and two were anti-EJ (3%) antibody positive. Other MSA we detected less frequently were the following: five anti-SRP (7%), three anti-Mi-2 (Mi2 alfa and Mi2 beta) (4%), and one anti-TIF1-ɣ (1%). In anti-Mi2-positive patients, two had ANA-positive speckled patterns at 1:640 and one cytoplasmic pattern (titer not available). No patient tested positive for anti-MDA5, anti-NXP2, or anti-SAE1.

Two patients resulted positive for multiple MSAs via LB: one was positive for anti-Jo1 and anti-SRP, and the other was positive for anti-Jo1 and anti-PL-7. Both patients had a diagnosis of ASyS: the first one (anti-Jo1 + antiSRP) had fever, fatigue, mechanic’s hands, arthritis, and myositis, confirmed via muscle biopsy and EMG. No features of necrotising myositis were found. The second one with anti-Jo1 and anti-PL-7 has incomplete features of ASyS, represented by ILD with dyspnoea, fatigue, and mechanic’s hands.

Isolated MSAs without an MAA were found in nine sera (13%). The most represented MAA was anti-Ro52, found in 32 patients (55%), followed by anti-PM/Scl (PM/Scl75 and/or PM/Scl100) in 10 patients (14%) and anti-Ku in 5 patients (7%). Unlike MSAs, the occurrence of an association between MSAs and one or more MAAs was more frequent (16 cases, 23%). Anti-Ro52 was found in 15 of these samples (93.7%), 3 samples were also positive for anti-PM/Scl (4%), and 1 was positive for anti-Ku (1%). In samples positive for anti-Ku, in 4/5 (80%), the ANA patterns were speckled, and one was negative. For anti-PM/Scl-positive samples, ANA patterns were speckled for 5/10 (50%) patients, homogeneous in 2/10 (20%) patients, two were negative (20%), and one was not available. Titer for the ANA results were not available retrospectively.

### 3.2. Autoantibodies via IP

IP detected MSAs and/or MAAs in 51 sera (74%), while an isolated MSA was found in 33 patients (48%).

The most frequent MSAs were anti-Jo1 in eight sera (12%), anti-EJ in three sera (4%), and anti-OJ in two sera (3%). The other MSAs found were nine samples with anti-NXP2 (13%), five samples with anti-MDA5 (7%), three samples with anti-SRP (4%), and three samples with anti-Mi-2 (4%). No samples tested positive for anti-PL7, anti-PL12, anti-TIF1-ɣ, or anti-SAE1. For anti-Mi2 samples, 2/3 (66.6%) exhibited an ANA pattern that was speckled, and for 1/3 (33.3%), results were not available.

Among MAAs, the most represented were anti-Ro52 autoantibodies with nine positive samples (13%), followed by anti-PM/Scl with four positive samples (6%) and anti-Ku with three positive samples (4%). All three anti-Ku-positive patients were ANA speckled. For anti-PM/Scl samples, the ANA pattern was speckled in 1/4 (25%) of the samples, and the pattern was nucleolar in 1/4 (25%) of the samples, speckled and nucleolar in 1/4 (25%) of the samples, and for 1/4 (25%) of the samples, a result was not available.

None of the samples were positive for more than one MSA, according to IP. By contrast, more than one MAA in the same patient was found with IP. In particular, two patients were positive for both anti-Ro and anti-Ku (3%). One of these patients was diagnosed as PM, but showing features of OM (such as sclerodactyly and Raynaud’s phenomenon); the other presented with an overlapped PM/SSc syndrome. A third patient had an association between MSA and MAA (anti-OJ and anti-Ro) with a diagnosis of OM (myositis and ILD) (Table 2). All three patients globally showed clinical features according with the MSA and MAA detected.

### 3.3. Comparison between LB and IP

MSAs and/or MAAs were globally detected with comparable rates via LB and IP methods as showed in Table 3. In particular, isolated MSAs were found more frequently using IP (48% vs. 13%; *p* < 0.0001), while multiple MSAs in the same sample were found only via LB but not with IP. LB and IP detected the same rate of anti-Jo1, anti-SRP, anti-EJ, anti-Mi-2, anti-OJ, anti-PM/Scl, and anti-Ku. Anti-NXP2 and anti-MDA5 were not found via LP, while IP detected them in 13% and 7% of sera, respectively (*p*: 0.003 and 0.058, respectively). By contrast, LB is more sensitive in detecting anti-Ro52 (43.4%) compared with IP (13%) (*p* < 0.0001).

### 3.4. Concordance between LB and IP

A moderate concordance rate between the two methods was found for anti-Mi-2 (k = 0.65) and anti-Jo1 (k = 0.574), and a substantial concordance was identified for anti-Ku (k = 0.73) and anti-PM/Scl (k = 0.72) antibodies. Double-positive patients for anti-Jo1 showed features typical of ASyS in most of the cases, namely arthritis, dyspnoea, and myositis, while Raynaud’s phenomenon and mechanic’s hands were found in 50% of cases. Patients with anti-Jo1 detected only via LB (five cases) or only via IP (two cases) did not differ in clinical diagnoses, though some differences were evident. The two patients that were anti-Jo1 positive only according to IP showed an ASyS diagnosis with fever, fatigue, myositis, arthritis, and dyspnoea. Five cases with anti-Jo1 detected only via LB showed additional less congruent symptoms, such as heliotrope rash (2/5), sclerodactyly (3/5), Gottron’s papules (3/5), calcinosis, and skin ulcerations (2/5). Two sera were anti-Mi-2 positive according to both IP and LB: both of the patients showed a diagnosis of DM, while the other two anti-Mi-2-positive patients only via LB or IP showed a PM diagnosis. DM-specific facial skin rash was found only in double-positive anti-Mi-2 patients, while fever, myositis, and arthritis were reported in all groups. Three sera were positive for anti-Ku according to both LB and IP: these patients showed a diagnosis of PM (1 case) and OM (2 cases). Fatigue (100%), sclerodactyly and puffy hands (two and one case), myositis (100%), Raynaud’s (100%), and dysphagia (75%) were the most common clinical features and symptoms associated with this specificity. Two patients were anti-Ku positive only via LB: both of them had a diagnosis of PM, with isolated myositis, arthritis, dyspnoea, and ILD only in one case. Although, considering the very few numbers of cases, the confirmed positivity for anti-Ku, via the two methods, included patients with a clinical picture more confident with the clinical diagnosis of OM. Two sera were anti-PM/Scl positive according to both LB and IP: these cases showed PM and OM diagnoses, with fatigue, myositis, Raynaud’s, and dysphagia. One patient was anti-PM/Scl positive only via LB and was diagnosed as OM, and two patients were positive only via IP and were diagnosed as PM and OM. Clinical features of these patients were similar and confident with anti-PM/Scl antibodies, with myositis, dyspnoea, dysphagia, and Raynaud’s. No atypical clinical features were detected in the three different groups of cases. Lower concordance was found for anti-EJ (k = 0.378), anti-SRP (k = 0.207), and anti-Ro52 (k = 0.17). Anti-EJ was found in one serum via both methods, in one serum only via LB, and in two sera only via IP: double-positive EJ serum achieved a diagnosis of DM, while the other three cases had a PM diagnosis: none of them showed features of ASyS. All four patients showed fatigue, myositis, dyspnoea, and ILD. No other symptoms were found. Anti-SRP was globally detected in five patients with LB (7.2%) and three patients in IP (4.3%). Anti-SRP antibodies were found in one serum via IP and LB, in a patient with a diagnosis of PM, characterised by myositis (muscle biopsy positive) and fatigue. Three sera were anti-SRP positive only via LB: these patients showed a diagnosis of DM and PM, with frequent cutaneous features, myositis, and skin ulceration, poorly fitting with the classical clinical pictures associated with anti-SRP. Two sera were positive only via IP, with diagnoses of PM and OM. Anti-Ro52 was the most frequent autoantibody found in LB, positive in 32 samples (46.4%), while it was found via IP only in nine sera (13.1%), with a poor concordance between methods. No peculiar clinical features were observed in anti-Ro52 positive cases, as expected. The concordance between the two methods could not be calculated for anti-PL-7, anti-PL-12, anti-OJ, anti-NXP2, anti-TIF1ɣ, or anti-MDA5, because no positive samples were found with one of the two methods. In particular, anti-OJ, anti-MDA5, and anti-NXP2 were found only via IP and not in LB, while anti-PL-7, anti-PL-12, and anti-TIF1ɣ were found only via LB, but not by IP. The two anti-PL-12-positive sera showed typical features of ASyS, while the two anti-PL-7-positive patients showed features of ASyS, with skin manifestations typical of DM. The patient with anti-TIF1gamma had a typical picture of DM (with concomitant cancer). Patients positive for anti-NXP2 (nine cases) and for anti-MDA5 (five cases) antibodies showed diagnosis of DM, with clinical features confident with the MSA detected.

## 4. Discussion

The diagnostic field of IIM remains a laboratory and clinical challenge; in fact, the correct detection of MSAs represents a primary tool to correctly classify patients and define their prognosis. Different automated multiparametric assays are currently widely used in routine workout. Nevertheless, the interpretation of the results still leads to some open points.

In this paper, we compared the performance of IP, as a reference method, and LB in the detection of MSAs and MAAs in 69 patients with a well-defined IIM diagnosis.

IP demonstrated higher sensitivity than LB in autoantibody detection, especially for isolated MSAs. Comparing the two methods, we did not find significant differences in the detection of anti-Jo1 and -Mi-2, both in terms of rate of positivity and concordance rate via Coehn’s k. Comparing the laboratory data with clinical features, we found a good correlation between anti-Mi-2 positivity and clinical picture, independently from the methods used for detection. By contrast, anti-Jo1-positive patients showed a complete concordance with ASyS and its features when detected via both methods and via isolated IP. The five patients that were anti-Jo1 positive only according to LB showed additional features typical of DM and not commonly observed in the clinical spectrum of ASyS. Nevertheless, anti-Jo1 detection via different methods seems to be comparable in terms of the associated clinical diagnosis. This observation is in line with a recent paper, describing a good agreement between a routine detection and centralised detection of anti-Jo1antibodies from a large cohort of ASyS patients [19]. Challenges persist for other non -Jo1 antibodies, such as anti-PL-7 or -OJ antibodies [19], as we found in our paper. In fact, we found anti-PL-7 and PL-12 only via LB (negative via IP). Anti-PL-12-positive sera showed concordant clinical diagnoses, while anti-PL-7-positive patients showed a wide spectrum of clinical manifestations, not perfectly confident with ASyS.

By contrast, anti-OJ antibodies were positive in two patients only via IP, not via LB. The detection of the complex OJ antigen has been elucidated by other authors [8,21], who developed a homemade ELISA with recombinant OJ to overcome the difficult interpretation via IP and the low sensitivity of LB [21,22], with different reactivity patterns described via IP for anti-OJ antibodies [23].

The rate of positivity of the single MSA via LB is comparable to what has been found by other authors [23], with anti-Jo1 detected as the first MSA, followed by SRP. Surprisingly, we did not find any sera positive for anti-NXP2 or anti-MDA5, or a sera that resulted positive via IP with a clinical picture suggestive of DM. These conflicting results have been previously observed by other authors. Anti-NXP2-positive DM is characterised by a juvenile onset with typical skin rash, calcinosis, dysphagia, myositis, and the absence of ILD [3,20,24]. These peculiar associations have been confirmed when anti-NXP2 antibodies have been detected via both IP and LB, while isolated LB positivity could detect anti-NXP2-positive patients with other clinical associations, such as cancer or IBM [25]. Anti-NXP2 antibodies and their DM pictures are, in our hands, confirmed only via IP, and surprisingly, none of the nine anti-NXP2-positive cases were detected via LB.

Moreover, anti-MDA5 antibodies were detected in five cases via IP, all with clinical features of DM and cutaneous ulcerations: to date, IP is considered to be the specific gold standard for detecting anti-MDA5 antibodies [26], while LB was not able to identify the five anti-MDA5-positive cases. More recently, other authors reported the high sensitivity and specificity of ELISA for anti-MDA5, especially in order to detect a quantitative value of this MSA [8,27,28], which is considered useful for the correlation between anti-MDA5 levels and clinical severity or response to treatment [28,29,30]. Another group developed a homemade immunofluorescence assay for anti-MDA5 detection, less expensive than ELISA, described as having high sensitivity [31].

Regarding MAA, a high concordance rate was found for anti-Ku and anti-PM/Scl antibodies. By contrast, anti-Ro52 antibodies were more frequently detected via LB compared with IP, also because protein-IP cannot detect Ro52 positivity unless a specific SDS-page gel is used, and for this reason, additional methods such as LB are considered more useful for anti-Ro52 antibody identification.

Finally, both LB and IP found the associations between MSAs and MAAs, with clinical pictures fitting with both autoantibodies. By contrast, LB detected two concomitant MSA (2.9%) not revealed via IP. Different authors previously described this phenomenon in different rates (8–11% of LB-positive cases) [7,31,32]. Ghirardello et al. recently reported how LIA and PMAT could give “false positives”, even in standard conditions, due to autoantibody polyclonality [23]. Several authors suggested that antibody titre quantification could be promising for this concern, to avoid the so called “clinical false sera” [33,34]. In particular, the clinical positive sera, i.e., sera confident with a specific IIM diagnosis, seem to be strongly positive antibodies, while MSA-positive sera (not associated with a specific clinical setting) showed more frequently a low titre and multiple positive MSAs/MAAs [34]. It is important to underline two major limitations to the present study: on one side, the small number of cases detected for some rare MAA and MSA specificities, which rises limitations in the statistical analysis; on the other side, the retrospective nature of this study, which is based on the results of laboratory methods performed in the period 2010–2022 characterised by different method settings and variables.

## 5. Conclusions

The LB method is used in clinical practice for the detection of different MSAs and MAAs in a short time and in a routine setting. However, the ability to detect MSAs and MAAs and the concordance with IP, the reference method, varies depending on the antibody sought, and the LB performance can be low for specific and rare MSAs and MAAs, as shown in our manuscript. In fact, we found a substantial concordance for anti-Ku and anti-PM/Scl between the IP and LB methods and a moderate concordance for anti-Jo1 and anti–Mi-2 antibodies. The concordance was fair for anti-EJ, anti-SRP, and anti-Ro52, while considering the small cohort, a correlation was not possible for anti-OJ (found only via IP), -PL-7, -PL-12, and -TIF1ɣ (detected only via LB).

IP showed very good performance for the detection of DM-associated MSAs, while LB poorly detected, in particular, anti-NXP2 and anti-MDA5. Both LB and IP showed good performance in the detection of anti-Mi-2-associated DM and ASyS-associated with anti-Jo1. Both LB and IP showed poor performance in the detection of other ASyS-associated MSAs, which are less frequent than anti-Jo1 antibodies.

Multiple MSAs were found only with LB in 2/69 sera while no cases of multiple MSAs were found via IP, as expected. LB showed a high number of positive sera for anti-Ro52 (46.4%) not confirmed via IP (13.1%), without specific associated clinical features. LB is a useful method for antibody detection that allows for the search of MSAs and MAAs in a clinical routine. However, the ability to detect autoantibodies is suboptimal for some MSAs/MAAs and is too sensitive for others, as compared with IP, with a discrepancy regarding the clinical picture. In cases showing a discrepancy between the clinical picture and the serology in LB, a confirmation test with IP could be helpful, even if the IP method may be time- and labour-consuming if used on large sets of samples.

## Figures and Tables

**Table 1 diagnostics-14-02192-t001:** Demographic features of the IIM cohort.

	ASyS, n = 11	Dm, n = 27	PM, n = 23	OM, n = 8
Female/Male ratio	8/3	18/9	19/4	6/2
Median age at onset, years (IQR)	43 (38.5–56.5)	43 (34.5–61)	47 (35–61.5)	39 (34.5–45)
Caucasian, n. (%)	10 (91)	26 (96)	11 (100)	7 (88)

IQR: interquartile range; ASyS: anti-synthetase syndrome; DM: dermatomyositis; OM: overlap myositis; PM: polymyositis.

**Table 2 diagnostics-14-02192-t002:** Clinical manifestation of patients with MSA/MAA association in IP.

	Patient GM	Patient PC	Patient ZL
Autoantibody positivity	Anti-Ro, anti-Ku	Anti-Ro, anti-Ku	Anti-OJ, anti-Ro
Clinical diagnosis	Overlap PM/SSc	PM	PM
Age at diagnosis, years	73	58	75
Clinical manifestations	Sclerodactyly Puffy hands Raynaud’s phenomenon Myositis	Sclerodactyly Puffy hands Raynaud’s phenomenon Myositis Dysphagia ILD	Myositis ILD

ILD: interstitial lung disease, PM: polymyositis, SSc: systemic sclerosis.

**Table 3 diagnostics-14-02192-t003:** Comparison between LB and IP in MSA and MAA detection.

	LB n. 69 (%)	IP n. 69 (%)	*p*-Value
Total positivity MSA/MAA	47 (68.1)	51 (73.9)	0.57
Isolated MSA	9 (13)	33 (47.8)	<0.0001
Double MSA	2 (2.9)	0	0.49
Jo1	11 (15.9)	8 (11.6)	0.6
PL-7	2 (2.9)	0	0.49
PL-12	2 (2.9)	0	0.49
EJ	2 (2.9)	3 (4.3)	1
OJ	0	2 (2.9)	0.49
SRP	5 (7.2)	3 (4.3)	0.71
Mi-2 α/β	3 (4.3)	3 (4.3)	1
NXP2	0	9 (13)	0.003
MDA5	0	5 (7)	0.058
TIF1gamma	1 (1.4)	0	1
Ro52	32 (43.4)	9 (13)	<0.0001
PM/Scl 75/100	10 (14.5)	4 (5.7)	0.157
Ku	5 (7.2)	3 (4.3)	0.7
SAE1	0	0	na

MAA: myositis-associated antibodies, MSA: myositis-specific antibodies, na: not applicable.

## Data Availability

The original contributions presented in this study are included in this article, further inquiries can be directed to the corresponding author.

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
