# Peer review of "Comparison of Lineblot and Immunoprecipitation Methods in the Detection of Myositis-Specific and Myositis-Associated Antibodies in Patients with Idiopathic Inflammatory Myopathies: Consistency with Clinical Diagnoses"

_diagnostics, 2024, doi:10.3390/diagnostics14192192_

Round 1

Reviewer 1 Report

Comments and Suggestions for Authors

The study by Angeli and colleagues is of significant interest to clinicians around the world using the LB by Euroimmune and who get negative results. Their data is very important

To my surprise, anti-MDA5 testing by LB was very disappointing in this study, which is an important message that authors should stress

Also, anti-NXP-2 testing by LB is very disappointing, and this is another message I would stress

Anti-OJ testing by LB was again confirmed to be disappointing

I was surprised to see that LB missed two anti-Jo-1 patients positive by IP with a compatible clinical scenario

It does not appear that anti-MI2 testing by LB misses true anti-Mi-2 and it does not appear in this study that anti-Ku testing by LB misses true anti-Ku either. I can't make a comment on anti-PM-Scl testing by LB in this study without having ANA patterns

I am confortable with the conclusions that IP has good performance for DM-associated MSA, but it is much more pertinent to stress that LB was awful, which needs to be said

It would have been very useful to have Hep-2 ANA results to discuss anti-anti-PM-Scl , and also anti- PM-Scl 75 and 100 results; for that reason, I do not agree with the conclusion that there is substantial concordance for anti-PM-Scl (LB = 10 and IP = 4)

Hep-2 results are also important for anti-Ku and anti-Mi-2. AC patterns would have been wonderful here

I would have preferred the authors to discuss anti-Ro-52 by LB versus IP, but I would try not to discuss anti-Ro-52 with another antibody. Having an anti-Ro-52 positive results with a positive antisynthetase result or a positive anti-Ku does not make me question the antisynthetase or anti-Ku result, so I would stick to the discordance between IP and LB for anti-Ro-52

As for seropositivity by LB in general in this study, I am not sure any firm conclusions in this study are being made

Can we trust an anti-SRP result alone , or do we trust it if a cytoplasmic staining on Hep-2 testing is there ?

Can we trust an anti-Mi-2 or anti-KU result alone, or de we trust if if a positive AC-4 or 5 is present ?

Can we trust an anti-PM-Scl result alone, or do we trust if a positive AC-8/9/10 is present ?

I like the paper but not the conclusions. I believe that LB misses anti-MDA5 and anti-NXP2 and rarely, anti-Jo-1, is such an important result

The clinical data here to referee LB versus IP is useful. Clinical data + Hep-2 results should be even more interesting

I believe the authors should stress the pitfalls of LB and not the advantages of their IP results, as we will not have access to their IP

I would just suggest the authors change their conclusions if they agree

Author Response

Point by point response to the reviewer comments

REVIEWER #1.

Comment #1. To my surprise, anti-MDA5 testing by LB was very disappointing in this study, which is an important message that authors should stress.

Also, anti-NXP-2 testing by LB is very disappointing, and this is another message I would stress.

Anti-OJ testing by LB was again confirmed to be disappointing.

I was surprised to see that LB missed two anti-Jo-1 patients positive by IP with a compatible clinical scenario.

It does not appear that anti-MI2 testing by LB misses true anti-Mi-2 and it does not appear in this study that anti-Ku testing by LB misses true anti-Ku either. I can't make a comment on anti-PM-Scl testing by LB in this study without having ANA patterns.

Response #1. We strongly agree with the reviewer’s comment, in particular for the low capacity of LB to identify anti-MDA5 and - NXP2 antibodies which are very important in the clinical setting to identify specific clinical features in myositis patients. Thus we added these points in the discussion paragraph in page 6 and 7.

Comment #2. I am confortable with the conclusions that IP has good performance for DM-associated MSA, but it is much more pertinent to stress that LB was awful, which needs to be said

Response #2. This important point was added in the conclusions section at page 7.

Comment #3. It would have been very useful to have Hep-2 ANA results to discuss anti-anti-PM-Scl , and also anti- PM-Scl 75 and 100 results; for that reason, I do not agree with the conclusion that there is substantial concordance for anti-PM-Scl (LB = 10 and IP = 4)

Hep-2 results are also important for anti-Ku and anti-Mi-2. AC patterns would have been wonderful here

Response #3.  We agree with the reviewer about the usefulness of having Hep-2 ANA results for all patients. However retrospective collection of the data didn’t allow us to have the data about ANA patterns for all patients. It is also to be considered that the ANA assessment was performed between 2010 and 2022 and we don’t have the AC patterns. However, we have the results of ANA in immunofluorescence (added in line 97-98 page 3) for the 3 patients positive for anti-Mi2; this information was added at line 140-141 page 3 and 168-196 page 4.

For the 5 patients with anti-Ku we added information available at line 156 page 4 and 172-174 page 4. For the PM/Scl sera data on ANA results available were added at line 156-160 page 4 and 172-174 page 4.

 We agree with the utility of having the data about Hep-2 ANA conducted and reported in a standardized way.

Comment #4. I would have preferred the authors to discuss anti-Ro-52 by LB versus IP, but I would try not to discuss anti-Ro-52 with another antibody. Having an anti-Ro-52 positive results with a positive antisynthetase result or a positive anti-Ku does not make me question the antisynthetase or anti-Ku result, so I would stick to the discordance between IP and LB for anti-Ro-52.

Response #4. Anti-Ro52 antibodies cannot be identified by IP unless a special gel is used to detect them, and for this reason other techniques (i.e. ELISA) are more commonly used with reliable results. We have specified this point in the Discussion section at page 7.

Comment #5. As for seropositivity by LB in general in this study, I am not sure any firm conclusions in this study are being made. Can we trust an anti-SRP result alone , or do we trust it if a cytoplasmic staining on Hep-2 testing is there ? Can we trust an anti-Mi-2 or anti-KU result alone, or de we trust if if a positive AC-4 or 5 is present ? Can we trust an anti-PM-Scl result alone, or do we trust if a positive AC-8/9/10 is present ?

I like the paper but not the conclusions. I believe that LB misses anti-MDA5 and anti-NXP2 and rarely, anti-Jo-1, is such an important result.

The clinical data here to referee LB versus IP is useful. Clinical data + Hep-2 results should be even more interesting

I believe the authors should stress the pitfalls of LB and not the advantages of their IP results, as we will not have access to their IP

I would just suggest the authors change their conclusions if they agree

Response #5. As correctly suggested by the reviewer, we have modified some points of the Discussion and the Conclusion section in order to be more detailed on the performance of LB more than on the comparison with IP, as we agree that in the future LB will be more available in laboratories worldwide rather than IP.

Reviewer 2 Report

Comments and Suggestions for Authors

In this manuscript the authors have compared immunoprecipitation (IP) and lineblot (LB) using the samples from sixty-nine patients with idiopathic inflammatory myopathies (IIM). The  detection of myositis-specific and myositis-associated antibodies (MSA-MAA) was performed. For this the authors have used 16 antigens: OJ, EJ, PL-12, PL-7, SRP, Jo1, Ro52, PM/Scl-75, PM/Scl-100, 99 Ku, SAE1, NXP2, MDA5, TIF1-gamma, Mi-2alpha, Mi-2beta as mentioned in the method section. The authors observed a substantial concordance for anti-Ku and anti-PM/Scl between the two methods. The authors came to the conclusion that IP is better than LB. The study is mostly complete. I have few minor comments

1.       The authors should provide a representative image of IP and LB for anti Ku and anti-PM/Scl.

2.       Since the IP is cumbersome for large number of samples as mentioned by the authors in the introduction section. The authors should discuss about it in the discussion section when proposing IP over LB.

3.       The lines 176 to 234 needs formatting as there are a lot of space between the sentences.

4.       The authors mentioned about 16 antigens in the method section but the Table 3 showed the result for 13 antigens.

Comments on the Quality of English Language

There are minor spelling mistakes in the text.

Author Response

REVIEWER #2.

Comment #1. The authors should provide a representative image of IP and LB for anti Ku and anti-PM/Scl.

Response #1. We certainly agree with the reviewer, but unfortunately, we have analyzed retrospective data in the decade 2010-2022 thus we do not have stored images of LB and IP films of the anti-Ku and -PM/Scl positive samples. The results were read during the test and reported in the clinical charts as results, without stored images.

Comment #2. Since the IP is cumbersome for large number of samples as mentioned by the authors in the introduction section. The authors should discuss about it in the discussion section when proposing IP over LB.

Response #2. We agree with the comment and we have added a sentence on this important point in the Conclusions section, page 8 lines 328-329.

Comment # 3. The lines 176 to 234 needs formatting as there are a lot of space between the sentences.

Response #3. Thank you for this observation, we have formatted the section indicated.

Comment #4.  The authors mentioned about 16 antigens in the method section but the Table 3 showed the result for 13 antigens.

Response #4. For anti-SAE1 we found no case in both LP and IP so it wasn’t reported in table 3. We specified this result at page 3 line 141-142 and page 4 line 167-168, and added in Table 3. 

Anit-Mi2 positive samples were Mi2 alfa and beta positive; we introduced this information at page 3, line 140.

Anti-Pm/Scl samples were considered positive if the sample tested positive for Pm/Scl75 and/or Pm/Scl100 in LB. We added this information at page 4, lines 152-153. 

Reviewer 3 Report

Comments and Suggestions for Authors

This manuscript assess the concordance of a commercial LB and home-74 made IP assays in the detection of MSA and MAA in  a well-established groups of patients with myositis. The importance of MSA and MAA detection for the diagnosis and risk assessment of patients with  IIM, have been highlighted in numerous studies hence raising awareness for differences between detection methods is of importance.

However this study has major limitations that should be stated in text.  

a.     the small cohort size (e.g. some differences which are of major importance did not reach statistical significance)

b.     the retrospective nature of this study - it should be clarified if the both tests were assessed at the same time as pre-analytic conditions may be of importance. 

Author Response

REVIEWER #3.

Comment #1. Major limitations:

- the small cohort size (e.g. some differences which are of major importance did not reach statistical significance)

- the retrospective nature of this study - it should be clarified if the both tests were assessed at the same time as pre-analytic conditions may be of importance.

Response #1. We agree with the reviewer comment and we introduced the study limitations in the Discussion section at page 7, lines 297-301.

Round 2

Reviewer 2 Report

Comments and Suggestions for Authors

In the revised version of the manuscript, the authors have addressed all my concerns. I support the publication of the revised manuscript.

Comments on the Quality of English Language

Minor editing in the language of the text is required.